# Viral Diagnosis of Hepatitis B and Delta: What We Know and What Is Still Required? Specific Focus on Low- and Middle-Income Countries

**DOI:** 10.3390/microorganisms10112096

**Published:** 2022-10-22

**Authors:** Amie Ceesay, Khaled Bouherrou, Boun Kim Tan, Maud Lemoine, Gibril Ndow, Barbara Testoni, Isabelle Chemin

**Affiliations:** 1INSERM U1052, CNRS UMR-5286, Cancer Research Center of Lyon (CRCL), 69008 Lyon, France; 2Medical Research Council Unit, The Gambia at London School of Hygiene and Tropical Medicine, Atlantic Boulevard, Fajara P.O. Box 273, Banjul, The Gambia; 3School of Arts and Sciences, University of The Gambia, Serrekunda P.O. Box 3530, Banjul, The Gambia; 4Department of Intensive Care Unit, Hôpital Lyon Sud, Hospices Civils de Lyon, 165, Chemin du Grand Revoyet, 69310 Pierre-Bénite, France; 5Division of Digestive Diseases, Section of Hepatology, Department of Metabolism, Digestion and Reproduction, Imperial College London, London W2 1NY, UK

**Keywords:** HBV, HDV, coinfection, diagnosis, serology, molecular, screening

## Abstract

To achieve the World Health Organization’s (WHO) goals of eradicating viral hepatitis globally by 2030, the regional prevalence and epidemiology of hepatitis B virus (HBV) and hepatitis delta virus (HDV) coinfection must be known in order to implement preventiveon and treatment strategies. HBV/HDV coinfection is considered the most severe form of vira l hepatitis due to it’s rapid progression towards cirrhosis, hepatocellular carcinoma, and liver-related death. The role of simplified diagnosticsis tools for screening and monitoring HBV/HDV-coinfected patients is crucial. Many sophisticated tools for diagnoses have been developed for detection of HBV alone as well as HBV/HDV coinfection. However, these advanced techniques are not widely available in low-income countries and there is no standardization for HDV detection assays, which are used for monitoring the response to antiviral therapy. More accessible and affordable alternative methods, such as rapid diagnostic tests (RDTs), are being developed and validated for equipment-free and specific detection of HBV and HDV. This review will provide some insight into both existing and diagnosis tools under development, their applicability in developing countries and how they could increase screening, patient monitoring and treatment eligibility.

## 1. Introduction

An estimate of the World Health Organization (WHO) considers viral hepatitis a public health issue, causing 1.5 million deaths per year. People living in low-income settings are faced with the hardest burden. Consequently, combating chronic viral hepatitis could improve global health equity [1].

Hepatitis B virus (HBV) and hepatitis delta virus (HDV) coinfection occurs on average in 10% of HBV chronically infected individuals. The acute HBV monoinfection may promote hepatic decompensation in a different way from acute coinfection, usually self-limiting in 90% of the cases [2,3]. More than 257 million people, or 3.2% of the world’s population, are estimated to be living with chronic hepatitis B infection (CHB). Among them, 12 to 15 million people have experienced HDV infections, with a varying geographical distribution of affected populations [4].

HDV/HBV coinfection has now been well demonstrated to worsen the course of chronic viral hepatitis by increasing liver damage and hepatocellular carcinoma, resulting in rapid progression to death (Figure 1) [5]. 

The prevalence of HBV/HDV coinfection has not been well documented throughout the world, especially in low- and middle-income countries, where HDV screening must be enhanced and optimized. Considering the higher risk of developing cirrhosis and liver cancer, the WHO optimistically recommends screening HDV in each patient chronically infected by HBV. There are two types of HDV/HBV coinfection, which are differentiated by the individual’s previous HBV infection status. Individuals with acute HBV infection exposed to HDV at the same time yield in HDV/HBV coinfection, while carriers of CHB subsequently exposed to HDV represent HDV/HBV superinfection. Acute HDV/HBV coinfection can occur as acute hepatitis with elevated transaminases and higher risk of fulminant hepatitis than acute hepatitis B monoinfection. This results in recovery in 95% of the cases while the remaining 5% may progress to chronic HDV/HBV coinfection [2,3]. The HDV/HBV superinfection is associated with increased risk of developing liver-related complications. Hepatitis B e antigen (HBeAg) is a soluble viral protein which is found early during acute hepatitis B and disappears soon after ALT peaks. In HBeAg-negative CHB, acute superinfection also results in the reduction in active HBV DNA synthesis in patients with low HBV DNA levels [6]. Superinfection leads to an acute flare-up of previously quiescent CHB. The rise in serum aspartate aminotransferase AST and alanine transaminase (ALT) may be the only indication of this event. Here, diagnosis is made by quantification of HDV RNA or serum antibodies of IgM anti-HDV and hepatitis B core IgG anti-HBc at the same time. Hence, the recommendation for screening all patients with CHB.

For decades, diagnosis of HDV/HBV coinfection has been based mostly on non-invasive techniques detecting serological, biochemical, and molecular markers. Detection of both HBV and HDV antigens and/or antibodies has been the diagnostic markers for serological assays such as the enzyme-linked immunosorbent assay (ELISA) and automated chemiluminescence immunoassays. Major international organizations of experts recommend that patients with chronic HDV/HBV coinfection should undertake periodic assessments of biochemical (serum levels of alanine aminotransferase, ALT), molecular (serum levels of HBV DNA and HDV RNA), and immunological (HBeAg, HBsAg, HDAg) markers (Table 1).

In this review, we described currently available tools for HDV and HBV diagnosis, to recommend simpler, accessible, and cost-effective screening techniques to apply in low- or middle-income countries.

## 2. Serological Markers and Related Diagnosis Tools

Serological markers of viral hepatitis infection allow the detection of the virus and the stratification of the disease phases. Infection phases produce various immunological responses, which result in varying serological markers (Figure 2). Serological screening is suitable for initial evaluation. The enzyme immunoassays and the automated chemiluminescence assay (CLIA) have been developed to target these markers and used as a first line of screening for viral hepatitis [7,8]. Several ELISA tests have recently been commercialized. ELISAs are antigen/antibody sandwich assays, which do not require well-trained laboratory personnel, advanced technologies, or high cost to be implemented in resource-limited countries These techniques target viral antigens and antibodies for instance hepatitis B surface (HBsAg), hepatitis B core-related (HBcrAg), and hepatitis B core antibodies (HBcAb), HBeAg) for HBV detection, and HDV antigens and antibodies (IgM and IgG anti-HDAg) [9,10].

### 2.1. HDV/HBV Coinfection First-Line Markers

Hepatitis B and delta coinfection is clinically indistinguishable from an acute icteric HBV infection. Thus, further evaluations are needed based on markers indicative of coinfection. First-line diagnosis of coinfection is based on simultaneous detection of different serological markers for both viruses. Markers to coinfection include HDAg, HDV RNA, and anti-HDAg (IgM and IgG) associated with HBV markers such as hepatitis B surface antigen (HBsAg) and hepatitis B core antibody (HBcAb). For HDV markers, IgM anti-HDV is detectable during the window phase of the infection, the period between the appearance of HDAg and the development of IgG anti-HDV (Figure 2). Then, IgG anti-HDV persists for several years after infection. Similarly, the diagnosis for HBV is based on HBsAg, anti-HBs and anti-HBc antibodies. ELISA and RDTs are the available basic laboratory techniques used for screening these markers.

Hepatitis B surface antigen (HBsAg), also known as the Australia antigen, was first isolated by Baruch S. Blumberg in the serum of an Australian Aboriginal person [11]. There are three types of surface antigens: large (L), middle (M), and small (S). These proteins form the viral envelope. Measurement of the HBsAg level has been widely adapted in clinical practice and clinical trials. Antibodies to HBsAg are important markers of immunity against HBV. This immunity is determined by the antibody levels in the bloodstream [12]. HDV is a satellite RNA virus that depends on HBV for propagation. It uses the HBsAg as a viral envelope and shares the same hepatocyte receptor for viral entry (Figure 3). However, the pathogenic interaction mechanism between HBV and HDV is not fully described.

### 2.2. Detection of HBsAg and Anti-HBs Antibodies

The first ELISA for HBsAg detection was developed by Hansson 1976 [13]. Since then, the measurement of the HBsAg, HBsAb levels and antibodies to HDV has been widely adapted in clinical practice. Several quantitative enzyme-linked immunosorbent assays have recently been commercialized: the Architect^®^ quantitative QT test (Abbott laboratory, North Chicago, Illinois, IL, USA), the Elecsys quantitative test (Roche laboratory, Basel, Switzerland), and the Liaison XL Murex^®^ quantitative test (DiaSorin laboratory, Saluggia, Italy). The first quantitative Architect test is the most widely used [14]. A head-to-head comparison of an Enhanced Chemiluminescence Immunoassay (ECLIA) named VITROS (Ortho Clinical Diagnostics, Raritan, NJ, USA), and Monolisa ELISA (Bio-Rad laboratories, Hercules, CA, USA) was performed by Tiwari et al. in 2020 for the detection of HBsAg in blood donors, in India [9]. Sensitivity of ECLIA and ELISA was 100%, while specificity was 99.97% and 99.7%, respectively. Therefore, either can be used for HBsAg screening. ECLIA has the advantage of being automated, but in terms of cost-effectiveness, ELISA is still the best candidate.

For the detection of anti-HBs antibodies (HBsAb), a previous study compared the performance of CLIA against ELISA [15]. Abbott Architect (CLIA) and Bio-Rad (ELISA) were compared by looking into concordance between the two test reports. CLIA showed a slightly better specificity and accuracy in detecting Anti-HBs, but analytical agreement was good between the two assays. Thus, these two immunoassays can be used in place of each other for detection of anti-HBs antibodies. However, in low- and middle-income countries, application of these techniques must be enhanced to reach a larger population.

### 2.3. Detection of Hepatitis B Core Antibody (HBcAb)

Hepatitis B core antibody (HBcAb) is an HBV-specific antibody that reflects the host immune response against HBV and can be used as a quantitative non-invasive marker of CHB. Hepatitis B core protein is the most immunogenic HBV antigen, and its antibody can persist for a long time. Anti-HBc is a classical serologic HBV marker that has been clinically used for more than 35 years [16]. HBcAb levels can be measured by immunoassays such as Architect Anti HBc II, (Abbott Laboratories, North Chicago, IL, USA) and Lumipulse Presto II (Fujirebio Inc., Tokyo, Japan) [16]. Kyo Izumida et al. developed a highly sensitive HBcAb assay suitable for clinical use [17]. It showed an excellent quantitative performance; the quantifiable range spans from 0.005 IU/mL to 1.500 IU/mL.

### 2.4. Detection of Anti-HDV Antibodies

For HDV, in 2018, Wranke and co evaluated a semiquantitative commercial kit for IgM anti HDV qualitative assay and found that patients with intermediate anti-HDV IgM levels had higher inflammatory activity than those with low levels or negative [18]. These findings suggest the crucial roles played by the different assays in categorizing HDV disease phases [19,20]. In the same year, DiaSorin has launched a CLIA (LIAISON XL Murex) (CLIA) for anti-HDV IgG and IgM screening. The reported turn-around time for the assay sensitivity and specificity was of 100% and 99.35%, respectively [10]. The CLIA was validated and compared with ETI-AB DELTAK-2 (DiaSorin, Saluggia, Italy), an anti-HDV ELISA, one year after its invention. The level of concordance between the reference method and CLIA was 97.7%. CLIA has also correctly detected low reactive samples not identified by the ELISA.

This and other techniques have revolutionized diagnosis of HDV and has been applied for routine screening in high-risk populations, as well as organ and tissue donors in the high- and middle-income countries.

### 2.5. Hepatitis Be Antigen (HBeAg) and Its Associated Antibodies (Anti-HBe)

According to the European Association for the Study of the Liver (EASL), chronic HBV infection natural history has been classified into five phases in relation to HBeAg, HBV DNA levels, alanine aminotransferase (ALT) values and eventually the presence or absence of liver inflammation [8]. These phases are characterized based on two major chronicity: infection vs. hepatitis. The phase 1 called HBeAg-positive chronic infection stage, is defined by the presence of serum HBeAg, very high levels of HBV and ALT persistence at a normal range as given by the conventional cut-off values (40 IU/mL). This phase is usually longer especially in patients infected perinatally and has been linked to preserved T cells functions at least until young adulthood [21]. Phase 2 known as HBeAg-positive chronic hepatitis B, characterized by serum HBeAg, high levels of HBV DNA and elevated ALT. This may occur after many years of the first phase and is more common and maybe rapidly reached in patients infected during adulthood. Even though the outcome of this stage is variable, many CHB can attain seroconversion and HBV DNA suppression and enter HBeAg-negative phase. Phase 3 known as HBV-negative chronic infection, is when the serum antibodies to HBeAg (anti-HBe) are undetectable or low, (2000 IU/mL), HBV DNA levels depending on individual patient (maybe low or high) and normal ALT according to the traditional cut-off values (UNL~40 IU/mL). Phase 4 is the HBeAg-negative chronic hepatitis do not have serum HBeAg but detectable antibodies to HBeAg (anti-HBe) and persistent or fluctuating moderate to high levels of serum HBV DNA and persistent variable values of ALT [22,23]. Phase 5 known as HBsAg-negative status or occult HBV infection is characterized by serum-negative HBsAg and positive antibodies to HBcAg (anti-HBc) with or without detectable anti-HBs, (Table 2). HBeAg is detected during acute infection. The disappearance of HBeAg, and especially the appearance of anti-HBe antibodies are signs of a favorable evolution and is associated with a significant decrease in viral replication. The HBe protein, which has a common amino acids (aa) sequence with the HBc protein, is secreted very early in the viral multiplication process.

HBeAg and anti-HBe detection are essential for the determination of the phases of CHB infection. It is associated with active HBV replication and increased risk for HCC [24]. It has been showed that HBeAg levels correlates directly with HBV DNA levels, hence, used as surrogate for HBV DNA levels [25,26]. HBeAg is also a risk marker for HBV mother to child transmission [27]. Even though HBeAg is an important indicator of CHB clinical management, immunoassays for detection of HBeAg have been reported to have poor results and low sensitivity and specificity [28]. This poor performance might be due to the possible cross-reaction between HBcAg and HBeAg. Indeed, HBeAg shares about 152 aa sequences with HBcAg, and most antibodies used in current HBeAg assays are not unreactive with HBcAg. Wang S-J et al. very recently developed a novel immunoassay using high affinity monoclonal antibodies (mAbs) recognizing NTR region unique to HBeAg [29]. Within these mAbs, 16D9 can detect the SKCLG (aa−10 to 5) motif on the N-terminal residues of HBeAg, which is absent in HBcAg. This target region makes it highly specific and sensitive in binding with another 14A7 mAb site of the HBeAg C-terminus such as STLPETTVVRRRGR, aa141 to 154. In contrast to widely used commercial assays of Abbott Architect and Roche Elecsys, the NTR-HBeAg completely eliminated the cross-reactivity with secreted HBcAg [29]. This method provides a robust tool to facilitate clinical diagnosis and drug development against HBV, avoiding possible false-positive detection attributed to the cross reactivity to circulating HBcAg [29]. To improve HBeAg detection techniques, Gani et al. has launched a multiplex assay based on giant magneto resistive (GMR) biosensor chips. This assay is quantitative platform for measurement of a panel of HBV serology markers (HBeAg, HBsAg, anti-HBs) [30]. This multiplexed detection system can provide preliminary screening and diagnosis of CHB infection and data in relation to the status, severity of disease and treatment monitoring. It is claimed to be simple and affordable to be applied in resource-limited countries.

In low- and middle-income countries, a combination of several diagnostic markers such as HBsAg, HBeAg, anti-HBs, and anti-HBe could be a good solution to facilitate access to a panel of serological testing.

### 2.6. Hepatitis B Core-Related Antigen (HBcrAg)

Hepatitis B core-related antigen (HBcrAg) is a useful therapeutic marker for CHB. HBcrAg is a composite biomarker that comprises HBcAg, hepatitis B e antigen and a truncated core-related protein p22cr. Serum HBcrAg has been reported to be a strong indicator of liver fibrosis [31]. Published data in Asia and European countries showed a strong correlation between HBcrAg levels and HBV DNA in treatment-naïve patients [31]. In addition, many studies found a concordance between serum HBcrAg and with both intrahepatic closed circular DNA (cccDNA) levels and its transcriptional activity [32,33,34]. In fact, serum surrogate markers reflecting the viral activity in the liver are urgently needed and the HBcrAg could be used to monitor liver disease progression. Recently, the fully automated CLEIA LUMIPULSE G600II (Fujirebio Inc., Tokyo, Japan) has been developed and used for HBcrAg quantification since its validation in 2014. Compared to quantitative HBV viral loads techniques, this immunoassay is cheaper (USD 15 vs. 60–200 per assay) and user-friendly.

In addition, a newer highly sensitive HBcrAg assay called iTACT-HBcrAg (Fujirebio Inc., Tokyo, Japan) has recently been launched and was considered as valuable for HBV diagnosis and monitoring [35,36]. In a comparative study between HBsAg and HBcrAg, serum levels of HBcrAg measured using iTACT-HBcrAg (8 times more sensitive than current HBcrAg assays) were significantly associated with HCC development [37]. A study reported its effectiveness in quantifying HBcrAg in HBeAg-negative patients under nucleoside analogues (NUCs) [38]. This method has the advantage of being easy to use, with a turn-around time of 30 min. Very recently, Shimakawa et al. developed a rapid diagnostic test based on immunochromatography to detect HBcrAg (HBcrAg-RDT) [39]. This simple and affordable tool was validated to identify highly viremic patients.

A non-invasive serological marker such as HBcrAg can be used as an alternative marker for HBV DNA quantification for chronic carriers’ management, guidelines for enrolment to treatment and monitoring, in low- and middle-income countries.

## 3. Rapid Detection Tests (RDTs)

Rapid detection tests (RDTs) have been developed and used for HBV screening while the development of RDTs targeting HDAg is still ongoing. RDTs are simplified versions of laboratory-based tests that have the potential to circumvent major barriers for people face with difficulty in accessing blood tests for hepatitis B and delta. In low- and middle-income countries, these types of tests could help reach a larger population during screening programs. The WHO recommends that an ideal RDT needs to meet the ASSURED criteria of being “affordable, sensitive, specific, user-friendly, rapid and robust, equipment-free and deliverable to end-users” [40]. RDTs have been developed for the detection of HBsAg and have the advantages to require no laboratory infrastructure, are easy to use, require little staff training, and provide a result within minutes [41]. The rapid detection of viral antigens such as HBsAg, anti HDAg by immunochromatography on membrane consists in depositing the sample to be tested (blood, serum, plasma and fingerpick blood samples) at one end of a nitrocellulose membrane. If the target antigen is present, it binds with a labelled antibody. Under the effect of a running buffer, the antigen-antibody complexes migrate by capillarity and are stopped by capture antibodies fixed on the membrane [42,43]. A positive result is indicated by the appearance of two-colored bands (control and test). These tests can be performed on serum or plasma, but also, on fingertip whole blood [44].

The sensitivity of RDTs varies considerably depending on the supplier and the matrix used, while their specificity is close to 100%. Some RDTs have shown excellent analytical performance for HBsAg detection, such as VIKIA HBsAg (BioMérieux, Marcy l′Etoile, France), Alere Determine HBsAg (Alere, Inc., Waltham, Massachusetts) and DRW-HBsAg v.2. (Diagnostics for the Real World, Ltd., Sunnyvale, California) [40,41]. These tests have shown to be equally sensitive (>98.5%) for the detection of various HBsAg mutants [45,46]. A recent meta-analysis evaluated the performance of 33 RDTs detecting HBsAg compared with a reference ELISA [47]. This study included 23,716 individuals: the overall sensitivity and specificity were 90.0% and 99.5%, respectively. Similarly, in July of this year a review paper on the performance of the Determine^®^ and other WHO-prequalified RDTs commonly used in many studies of CHB in both developed and developing worlds was published [40,48]. The analytical performance of the tests was not influenced by the type of matrix tested (serum, plasma, venous or capillary whole blood).

HBV RDTs have been a critical tool in screening HBsAg seroprevalence, and it is imperative to developpe RDTs targeting anti-HDAg. Thus, RDTs for HDV could be the ideal solution to screen HDV in HBV-positive patients widely, thanks to its cost-effectiveness and simple application.

## 4. Biochemical Markers and Related Diagnosis Tools

### 4.1. Aspartate Aminotransferase (AST)/Alanine Aminotransferase (ALT)

The assessment of the severity of liver disease is important to identify patients for treatment and HBV infection surveillance. Biochemical parameters, aspartate aminotransferase (AST) and alanine aminotransferase (ALT), are indicative of inflammation and liver disease. Serum ALT and AST is the cheapest and most widely used laboratory parameter for the evaluation and follow-up of chronic HBV-infected subjects. Measurement of ALT and AST are used for monitoring HBV chronic carriers’ progression to liver disease [49]. In addition, ALT levels, associated with HBV DNA and HBeAg quantification, can help classifying HBV chronical phases [8]. Serum levels of ALT and AST can be measured using different device such as the Hitachi Automatic Analyzer 7600 (Hitachi, Tokyo, Japan). Their levels are considered to be elevated when it is higher than 40 IU/L, according to the conventional value [8,50]. L-W Song et al., showed a correlation between HBcAb and ALT levels [51]. A multicenter study analyzed 1940 HBV-related HCC patients who underwent hepatectomy and classified them based on baseline HBV DNA load and AST/ALT ratio. This classification stratified patients with distinguishable prognoses after hepatectomy [52].

Although a slight increase in ALT levels is an indicator of liver damage, the contrary has been reported in African HBV chronic carriers. In Senegal, patients with normal ALT values, had high liver stiffness measurement and viral load, ranged between 7 to 13 kPa and 3.2 log_10_ to 4.2 log_10_ IU/mL, respectively. These results reflect active infection and significant liver fibrosis [53]. Another study on a Chinese cohort also showed that some patients with normal ALT exhibited significant necroinflammation and fibrosis in liver biopsy. This discordance might be due to an intermittent small-scale liver injury along with chronic infection [49]. Importantly, such patients face a high risk of cirrhosis and other end stage clinical complications without treating liver injury.

A study developed and validated a diagnostic prediction score for treatment eligibility in African individuals with chronic HBV infection [54]. This simple score named TREAT-B (Treatment eligibility in Africa for hepatitis B virus), based on HBeAg and ALT, had a high diagnostic accuracy for the selection of patients for HBV treatment. This score could be useful in low- and middle-income settings where HBV DNA quantification is not routinely available. Based on this score a recent report evaluated the validity of an algorithm selecting HBeAg-positive and HBeAg-negative women with ALT as a predictor of high HBV DNA level [55]. Another study used TREAT-B criteria of HBeAg and ALT levels, linked to socio-demographic data, to specify treatment eligibility [56]. In this study, TREAT-B cut-offs (≥2) determined more than half of participants qualified for treatment, from low endemic regions compared to those from high endemic regions. Alternatively, other markers such as the gamma-glutamyl transpeptidase (GGT) to platelet ratio (GPR) have been reported to be an independent predictor of significant fibrosis in a Gambian cohort [57].

### 4.2. Alpha-Fetoprotein (AFP)

The level of alpha-fetoprotein (AFP) was the most common biological marker used for the clinical diagnosis of HBV-related hepatocellular carcinoma (HCC). In addition, the combination of ultrasound and AFP has been shown to provide some additional detection of 6–8% of cases compared to ultrasound alone [58]. However, it is limited with relatively low sensitivity and high false positivity in HBV-related HCC. Indeed, AFP cannot distinguish between small HCC masses and liver cirrhosis. Consequently, the EASL and the American Association for the Study of Liver Diseases (AASLD) recommend the use of AFP in combination with ultrasound and other biomarkers for HBV-related HCC diagnosis [59]. In fact, AFP is still among the best biomarkers for HBV-related HCC prediction and recurrence [60,61]. A combination study, which measured serum levels of des-carboxyprothrombin (DCP) with AFP (LIMPULSE G1200, Fujirebio Inc., Tokyo, Japan), suggested that the association of AFP with DCP might be the best and most cost-effective strategy for monitoring patients for HCC [62]. Similarly, AFP plus interleukine-34 (IL-34) showed high sensitivity and specificity in reliably predicting the development of HBV-related HCC among CHB in China [63].

## 5. Molecular Markers and Related Diagnosis Tools

In the case of HBV/HDV coinfection, serological screening is the first line of diagnosis, which is followed by HBV DNA and HDV RNA detection and quantification. It is crucial to detect and quantify HBV DNA and HDV RNA for differentiation of active and resolved infection, and to also determine treatment eligibility.

### 5.1. HBV and HDV Viral Load

Quantification of HBV DNA and HDV RNA in blood has become a critical tool in the assessment and management of chronic coinfection. Historically, hybridization assays were used to estimate viral loads in serum or plasma samples, but their level of sensitivity was suboptimal. This was followed by the application of a polymerase chain reaction (PCR) test, a highly sensitive test for nucleic acids to HBV DNA and HDV RNA detection. For decades, several laboratory evaluations of real-time PCR assays for hepatitis B and delta nucleic acid quantification have shown the importance of these techniques for diagnosis and prognosis of HBV/HDV coinfection.

The HBV genomic DNA is a relaxed-circular DNA (rcDNA) of approximately 3.2 kb in length with a complete minus strand and an incomplete plus strand. The viral genome encodes four overlapping open reading frames (ORFs), C, P, S, and X, from which functional viral proteins are produced. rcDNA is converted into covalently closed circular DNA (cccDNA) in infected cells. Its replication triggers immune responses, which leads to liver damage. It is important to determine the amount of HBV DNA for a better understanding of the natural course of the disease. It can be used to determine different phases of chronic HBV infection. Viremia, ALT, and histological outcome constitute major components in management and decision to initiate treatment in HBV chronic carriers. The monitoring of HBV DNA in serum is as important as serological markers in predicting the clinical outcome of the infection.

### 5.2. Detection and Quantification of HBV DNA

The COBAS Ampliprep/TaqMan HBV test was the first commercial assay for the detection of HBV viremia [64]. This system was validated for the detection of HBV genotypes A to G with a sensitivity in the range of 4–12 IU·mL^−1^. It is now commonly used, and its efficiency is undeniable. Several real-time PCR-based HBV DNA quantification assays have also been developed with an increased sensitivity, such as the Abbott Real-Time HBV Quantification kit (Abbott Laboratories, Abbott Park, IL, USA), the Versant HBV DNA Assay 3.0 (bDNA-Siemens Medical Solutions Diagnostics, Tarrytown, New York, NY, USA) and the Amplicor HBV Monitor Test (Roche, Alameda, CA, USA) [65,66]. However, due to their high cost, they remain unavailable and inaccessible to developing countries.

Compared to COBAS Taqman Realtime PCR and Abbott Realtime HBV assay, in-house real-time PCRs for HBV DNA quantification were developed. A low-cost in-house quantitative real-time PCR assay was validated to monitor HBV viral load in Gambia and Senegal by S Ghosh et al. in 2016 [67]. It was developed at the INSERM research unit U1052, in Lyon, France. This syber-green in-house assay was revealed to be highly sensitive, in detecting viral loads of 5 IU·mL^−1^. When compared to COBAS Taqman and Abbott, this assay is cheap, accurate, simple to use and it can detect major HBV genotypes. In addition, even though the assay does not detect all HBV genotypes, it can only detect the major genotypes that are reported in developing regions such as Africa (genotypes A, B, C, D, E and F) [68].

### 5.3. New Tools under Development for Diagnosis of HBV/HDV Coinfection

Most recently, newer, more sensitive, and automated commercial assays have been launched. The droplet digital PCR (ddPCR, Bio-Rad laboratories, Hercules, CA, USA), the Aptima HBV Quant assay (Hologic, Marlborough, MA, USA), GeneXpert HBV viral load (Cepheid, Sunnyvale, CA, USA), and loop-mediated isothermal amplification (LAMP, New England Biolabs, Ipswich, MA, USA) are claimed to be rapid, sensitive, specific, and highly versatile assays. Aptima has commonly been used in Europe, while LAMP is reported to be cost-effective and may be a new solution for low- or middle-income countries, where many fatal infectious diseases are endemic. HBV LAMP is an attractive alternative to PCR, but it requires further field evaluations [69]. Finally, GeneXpert is fast, easy to use, and facilitates the decentralization of management of CHB [70,71,72]. An evaluation of the GeneXpert HBV viral load assay has been performed by Florence Abravanel et al. in 2020 [73]. The analytical sensitivity of the GeneXpert assay was reported to be excellent, with a lower limit of detection (LOD) in keeping with the manufacturer’s stated value (10 IU/mL). GeneXpert HBV viral load assay can be revolutionized testing in low-income countries with an existing Cepheid GeneXpert system for Human Immunodeficiency Virus (HIV), hepatitis C virus (HCV) or tuberculosis (TB). Consequently, only the GeneXpert HBV cartridge will then be required for monitoring and treatment enrolment of HBV chronic carriers.

### 5.4. Detection and Quantification of HDV RNA

Efforts to improve the diagnosis of HDV RNA started in 1988 with the development of a non-invasive HDV RNA hybridization assay (dot blot) that has a sensitivity of 80 to 85% [74]. A more sensitive polymerase chain reaction (PCR) test was claimed to show great potential for diagnosis and monitoring of chronic hepatitis delta [75]. Despite its high sensitivity, this test did not detect all genotypes due to the higher variability of this pathogen.

The development of technologies capable of quantifying HDV RNA and identifying various genotypes in the serum of patients in the most efficient and standardized way is essential [76,77]. Assays such as in-house quantitative PCR (qPCR) and automated one-step quantitative reverse transcription PCR (RT qPCR) tests have been developed for HDV viral load and genotyping in Europe [78]. In the same year, Ferns and al. developed a standardized real-time RT-qPCR test, which uses a full-length genomic RNA transcript to minimize the risk of producing false-negative or underestimation of HDV viral load [79].

Currently, several laboratories have been developing and improving an in-house RT qPCR and commercial assays to determine HDV viral load by using sera or plasma from patients infected with all HDV genotypes [80]. The last decade has experienced several efforts to establish a standardized HDV RNA viral load measurement tools and assays. The introduction of an in-house PCR, a type of assay developed and validated in some laboratories has contributed to the quest for more reliable, sensitive, and specific assays for HDV viral viremia. It has been widely used for decades to quantify HDV RNA, involving several nucleic acid amplification tests with internal standards of different origins [76,78,81,82]. These assays are easy to perform and have been useful for several HDV genotypes [78,81]. To our knowledge, several in-house assays [79,83,84,85,86], with easy-to-apply and inexpensive protocols, have not been evaluated on a larger panel of clinical samples of different genotypes in resource-limited countries. In addition, despite several improvements to increase their sensitivity and specificity for HDV diagnosis, some recent studies have reported poor performance, underestimation, or failure to detect and quantify positive HDV RNA samples. There is therefore a need to further validate these assays with many cohorts, particularly in Africa and South America.

Overall, these techniques for quantifying both viruses can be used in the context of HBV/HDV coinfection. Evidence has indicated that HDAg down regulates HBV replication by repressing the activity of both HBV enhancer regions. In relation, studies have reported significant and persistent higher HDV levels in HBV/HDV carriers compared to HBV viral load [87,88].

### 5.5. HDV Genotyping

Hepatitis delta isolates have been classified into eight major genotypes distributed over different geographical region. Molecular techniques have been the backbone of their discoveries [89]. Between the genotypes, HDV have a sequence variation ranged from 11 to 19% heterogeneity [2,90,91]. In the past, Wu et al. developed a genotyping method based on restriction fragment length polymorphism (PCR-RFLP), a variation conventional PCR in which amplicons are treated with restriction enzyme and the blueprint of the genotypes are viewed on a gel image [92].

Currently, RT-nested PCR, which has been the gold-standard diagnostic technique for occult hepatitis B infection, is now used for genotyping HDV genotypes [2,93]. Nested PCR is a modification of PCR protocol to improve sensitivity and specificity. The first step of HDV RNA genotyping involves converting RNA to a cDNA using Reverse transcription method. The principle of nested PCR includes the application of two pairs of primers and two steps of PCR reaction, where the first step of primers is designed to target sequences upstream from the second pairs of primers and used in an initial PCR reaction. The PCR products from the first PCR are applied in the second PCR as a template for the second set of primers and a second [94].

Studies validated both nested and restricted-fragment-length polymorphism (RFLP)-PCR in genotyping HDV isolates to identify HDV genotype one and results obtained have been reliable [2,93]. A study in 2016 used semi-nested RT-PCR method to detect HDV isolates from Sudanese blood donors with HBV/HDV coinfection. The method has proven to be sensitive and specific in identifying HDV/HBV coinfection in that population [95].

### 5.6. HBV Pre-Genomic RNA

Circulating HBV RNA was first observed in 1984 in the form of HBV DNA–RNA hybrid molecules [96], while HBV RNA was first detected in 1996 in the serum of infected patients by Köck et al. [97]. To date, much data have been accumulated for the biomarker [98]; to this end, we concentrate our focus on the existing assays and techniques used for its quantification. During HBV infection, virions bind to the cell surface (hepatocytes), and nucleocapsids are shuttled to the nucleus where they release rcDNA into the nucleus. rcDNA then converts to cccDNA which acts as a mini chromosome and a template for the transcription of all viral RNAs. cccDNA produces different types of RNA transcripts of various sizes: a greater-than-genome pre-genomic RNA (pgRNA) and sub-genomic RNAs that have different 5′ends. cccDNA is considered as the hallmark of HBV infection. HBV infection cannot be eliminated due to the persistence of cccDNA in the nuclei of infected hepatocytes [99]. However, it can only be measured by using invasive liver biopsy. It is imperative to develop non-invasive surrogate markers to monitor the quantity and/or transcriptional activity of cccDNA [34].

The full characterization of the circulating RNAs is ongoing. Professor Lu Fengmin’s team has determined that peripheral blood HBV RNA is prevalently constituted by pgRNA in 2016 [100]. It is released into the serum in the form of enveloped pgRNA containing virions. Several studies have shown the potential use of HBV RNA as a surrogate biomarker to cccDNA [34,101,102]. Moreover, HBV RNA levels have been shown to correlate with other HBV markers such as HBV DNA, HBsAg, HbcrAg [103,104,105].

Although HBV RNA quantification has not been reported for all HBV genotypes, previous studies have highlighted that HBV genotypes may influence serum HBV RNA levels [106]. In addition, serum HBV RNA levels and HBV RNA/DNA ratios change significantly during chronic infection [100]. Similarly, this also supports the theoretical idea that HBV RNA is a good predictor of cccDNA levels but depends on HbeAg status. It is widely reported that HBV RNA levels distinguish the HBV chronic phases. In a cohort of 135 Chinese patients with chronic hepatitis B, 77 were identified as inactive carriers with significantly low RNA levels, while the remaining patients were HbeAg negative, with higher RNA levels [101].

During treatment of HBV patients with NUCs and pegylated interferon alpha (Peg-IFN-α), HBV RNA has the potential to predict HbeAg seroconversion [102]. HbeAg seroconversion is considered as a precondition for HbsAg loss or seroconversion, both of which represent stable remission of HBV infection to reach treatment endpoint. While HbeAg seroconversion is a result of treatment remission, makers such as HBV DNA, ALT, and HbsAg can be used to predict disease activity. In addition to HBV DNA, HBV RNA represents another serum marker of treatment endpoint. In a cohort of HbeAg-positive patients, Van Bommel et al. found that patients on NUCs and peg-IFN treatments who achieved seroconversion had a greater decline in HBV RNA levels than patients who did not [103]. In another study, HbeAg-positive patients treated by NUCs for 152 weeks, had negative HBV RNA serum levels, and HbeAg loss [107].

Since the discovery of HBV RNA marker, different attempts have been made to develop techniques to quantify it in liver cells and patients’ serum. Three techniques have mainly been developed: RACE-PCR (Rapid amplification of cDNA-ends PCR), standard RT-qPCR (Reverse transcriptase PCR), and ddPCR (droplet digital PCR). RACE PCR targets a short pre-genomic HBV RNA sequence at the primer 3′ end. It was first developed by Köck et al. [97]. Van Bommel et al. modified RACE techniques were cDNA created is amplified and detected by real-time PCR (RT-qPCR) to allow relative quantification of HBV RNA based on full length (flRNA) and truncated (trRNA) [102,103]. Many research teams have also adapted an RT-qPCR approach to detect and quantify total serum RNA. Here, reverse transcription involves a primer binding to the 5′end of HBV RNA. In addition, a sensitive method called RT-ddPCR (reverse transcriptase droplet digital PCR) have been used for quantification of HBV RNA [104,108]. Wang et al. first used this method with amplifying primers within PreC/C and polymerase coding regions and had poor LOD. Similarly, Limothai et al. used this technique to detect serum HBV RNA and reported an improved LOD [108]. The development of commercial HBV RNA kits is on the pipeline. In 2018, Abbott laboratories have launched an automated assay for quantification of HBV pgRNA using a dual-target qRT PCR approach on the Abbott m2000 sp/rt system. This automated assay had high sensitivity in quantifying RNA [109]. However, this available test has not been standardized yet.

In early 2022, a new automated prototype quantitative HBV RNA assay referred to as an investigational cobas^®^ HBV RNA assay was developed. It aimed to quantify circular HBV RNA (cirB-RNA) by targeting the 3′ end of HBV transcripts, allowing it to detect HBV viral RNAs expressed from cccDNA [110]. The cobas^®^ 6800/8800 system (cobas HBV RNA, Rohe Diagnostics, Pleasanton, California, CA, USA) was validated to determine its analytical and clinical performances with both EDTA plasma and serum samples from 36 treatment-naïve patient to amplify different cirB-RNA genotypes (A, B, C, D and E). The technique produced high sensitivity, precision and specificity when compared to HBV DNA viral load of cobas HBV DNA assay [110]. Even though cobas HBV RNA has been calibrated in units of copies/mL, where one copy of RNA is representative of an international unit for DNA molecules, it is not approved for clinical use by any regulatory body. Therefore, further studies with large cohorts of CHB and clinical trials (drug) are recommended to validate cobas HBV RNA as a tool for assessment of the clinical relevance of cirB-RNA biomarker.

HBV RNA in serum is an interesting biomarker, representing cccDNA levels and transcriptional activity, at least in HBeAg-positive patients. Thus, this marker can allow diagnosis and monitoring among this group of patients. However, in the context of treatment response management, there is a need for more studies involving other HBV biomarkers in combination with HBV RNA. Finally, these techniques are currently difficult to apply in low- and middle-income countries, due to their high cost and the technological complexity of their application.

### 5.7. A Point-of-Care Test (POCT) for HDV Anti-IgM and IgG

Since the discovery of hepatitis delta more than four decades ago, there is no single WHO prequalified point-of-care testing tool. This might be due to the existing assays, which include an enzyme ELISA or molecular techniques for HDV RNA viral load and genotyping. These assays are quite expensive, require laboratory equipment and trained personnel and takes several hours of preparation to obtain result. The immunosorbent assays, which were developed by advanced world and are routinely used to screen anti-HDV. For years, it has been working well with better sensitivity and specificity and with wide acceptance around the world until recently, when the WHO made a timeline to eliminate viral hepatitis by 2030, with the aim to screen people in the poorest regions of the world, where there is limited access to hospitals or advanced laboratory. Thus, a POCT is an ideal test for HDV defection since it is cost-effective and easy to use. It could increase HDV screening in clinics, hospitals, communities in hard-to-reach areas in many countries. Therefore, it has been among the most efficient and easy to use tools for screening people with other viral hepatitis. Studies have conducted community and population-based studies to identify hepatitis B surface antigen-positive individuals in resource-limited countries such as the Gambia [111,112].

Efforts have been made to develop a POCT for detecting anti-HDAg and more is needed for simplification of HDV diagnostics to attend the World Health Organization target to eliminate viral hepatitis by 2030, especially for resource-limited countries. In 2019, a collaborative study was carried out by some laboratories from three countries, Germany, France, and Turkey to develop a rapid pan-genotypic POCT for anti-HDAg detection [113]. The proposed POC strip will allow fast and reliable detection of all eight HDV genotypes. It was first validated with a panel of 332 HDV-positive and 142 HDV-negative patients’ samples and all genotypes. It yielded a promising calculated sensitivity of 94.6% and a specificity of 100%. However, the kit is yet to be deployed in clinics, hospitals, and communities even after successful validation. The POC prototype otherwise known as the pan-genotypic lateral flow assay (HDV rapid test) was used to screen 4103 HbsAg-positive and 166 HbsAg-negative and anti-HCV-positive sera from China and Germany. Compared to a commercial ELISA, the HDV rapid test has a sensitivity of 94.6% and a specificity of 100%. In Germany, HbsAg- and anti-HBc-negative samples from HCV patients were tested with an in-house ELISA using the same recombinant hepatitis delta antigen (HDAg) [114].

To the best of our knowledge, this is the first study that has attempted to develop a anti-HDAg POC test after more than four decades of HDV infection discovery, i.e., since the 1970s. The elimination of viral hepatitis should target people worldwide, even those in rural or difficult-to-reach and resource-limited settings. This is the right point to venture into strategies to diagnosis all, and this would involve developing simple and efficient diagnostic tools such as point-of-care tools to screen hepatitis B surface-positive individuals as recommended by the American Association for the Study of Liver Diseases (AASLD) [115].

## 6. Summary

HBV and HDV testing in resource-limited countries could be more strategized to focus on the molecular POCs, RDTs and some relevant biochemical assays, which are cost-effective and could be readily available in many developing countries. For instance, the Xpert platform (molecular point of care) is among the tools with promising potential particularly for countries with existing GeneXpert platforms, being routinely used for screening Mycobacterium tuberculosis (MTB), (HIV), (HCV), and Xpress CoV-2 diagnosis among other pathogens. The GeneXpert systems currently available include GeneXpert II, IV, XVI and Infinity. To utilize these technologies for HBV DNA viral load, all that is required is system upgrading and this makes it cheaper and readily available for the developing settings. In addition, considering the genetic diversity of HBV, the system, though still under development, has been reported to detect all HBV genotypes and subtypes. Perhaps soon Cepheid could develop and incorporate HDV RNA cartridge in this platform.

Similarly, the LAMP techniques recently validated in Senegal have improved performance compared to when they were introduced. They are also already available in some African countries for other pathogens including malaria. In Northern Namibia, LAMP techniques have been used for screening in low-malaria transmission settings, where RDTs for malaria have proven unreliable.

Given the general knowledge that people infected with HBV at birth or the first five years of life contributes significantly to HBV-related death statistic worldwide, the WHO in 2017 recommended the use of HbeAg testing for pregnant women, where HBV DNA testing is not available, to determine treatment eligibility for tenofovir prophylaxis in order to prevent HBV mother-to-child transmission. In this context, the improved version of HbeAg assay previously described as a serological tool could become handy for middle- and low-income countries (Table 3).

Another simple and low-cost diagnostic assay that could help tackle the issue of access to diagnostic tools to assess treatment eligibility in low- and middle-income countries is HbeAg in combination with ALT. The simple score diagnostic tool (Treat-B) is based on HbeAg and ALT levels and is in development by the PROLIFICA team in the Gambia (West Africa) for HBV treatment eligibility. The Treat-B tool, without the need for testing HBV DNA viral load, proved useful for selecting patients for HBV treatment in African settings.

Hepatitis delta, commonly referred as the “forgotten disease”, has been around for more than four decades, due to the lack of awareness among some health care workers and the general public. With more than 20 to 60 million people infected, several efforts have been made to improve and simplify its diagnostic techniques for both developed and developing settings. Considering the complex genome of the virus, some of the developments are rather slow. Since its discovery, ELISAs, and molecular techniques such as one-step qPCR, many in-house assays, nested PCR and, most recently, anti-HDAg prototype POC have been employed. ELSIAs, considered the diagnostic gold standard for HDV, are cheaper and easily accessible even for those in resource-limited settings. In addition, a nested PCR assay, which is widely available for detection of other pathogens such as malaria in some African countries, could be effective in screening HBV/HDV coinfections among HbsAg-positive individuals. However, some HDV RNA techniques, which are highly sensitive and specific, are more expensive and require trained personnel, and dedicated equipment may not be available in the developing world. In a nutshell, an anti-HDAg (in progress) test could make an impact in HBV/HDV coinfection seroprevalence in low- and middle-income countries since RDTs have been used for diagnosis of many pathogens in difficult-to-reach regions in the world (Table 3).

## 7. Conclusions

We reviewed all known HBV and HDV markers used for diagnosis of acute and chronic hepatitis infections. Serological marker screening, followed by biochemical and molecular marker quantification, is still the ideal protocol for hepatitis diagnosis. Current diagnostic tools related to these markers, to detect HBV/HDV coinfection generally show appreciable performances. However, considering the higher risk of developing liver disease and HCC in the case of coinfection, improved precision and quantification in screening HBV and HDV are mandatory. Some markers such as HDV RNA and HBV DNA seem reliable for estimation of viral load, although their detection and quantification are compromised due to high detection limits and costs. The development of new diagnostic tools is still in progress and is hoped to be standardized in the future.

To achieve the WHO goals of eradicating viral hepatitis globally by 2030, the regional prevalence and epidemiology of both viruses must be known to implement strategies to mitigate risk factors and reduce transmission. To increase screening and access to treatment for viral hepatitis, particularly in resource-limited countries, it is imperative to adopt new screening approaches—where possible, non-invasive, ideally using capillary blood, oral fluids, self-testing, mobile testing, and multi-disease testing. Rapid tests are very promising in this sense and their potential usefulness is particularly important for resource-limited countries or for fragile populations. Innovative technologies that detect viral hepatitis markers in sweat, for example, could be very useful, especially in tropical areas where high temperatures favor profuse sweating. Proposals for innovative technologies are underway in many directions.

The adoption of RDTs to screen and test areas in resource-limited countries, where HDV and HBV are highly endemic, will facilitate diagnosis and improve monitoring of serological and virological markers of infection. This is important for better referral to treatment and care, thus contributing to the elimination of viral hepatitis.

## Figures and Tables

**Figure 1 microorganisms-10-02096-f001:**
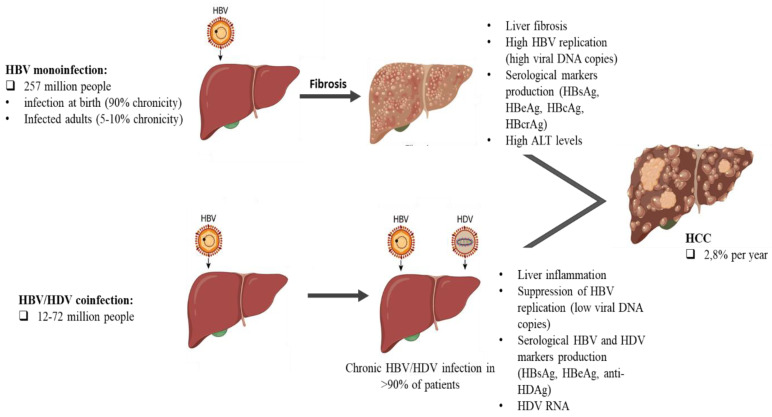
Representation of HBV monoinfection and HBV/HDV coinfection disease stages.

**Figure 2 microorganisms-10-02096-f002:**
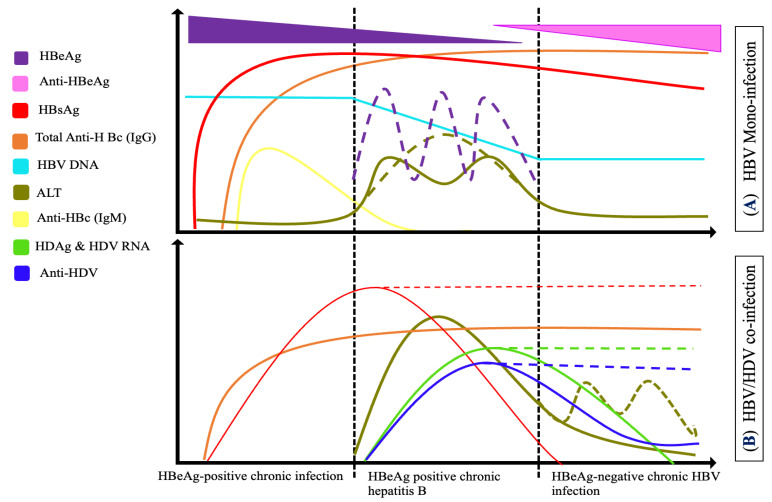
Serological markers of HBV monoinfection (**A**) and HBV/HDV coinfection (**B**): the diagnosis of HBV monoinfection and HBV/HDV coinfection is based on simultaneous detection of different serological markers. The markers of chronic HBV and acute HDV include HBsAg, HBeAg, HBV DNA, anti-HDAg, HDV RNA and immunoglobulines M and G. The HDV markers are present only transiently and disappear during early remission. Antibodies to HDV also disappear with time in acute resolving HDV infection.

**Figure 3 microorganisms-10-02096-f003:**
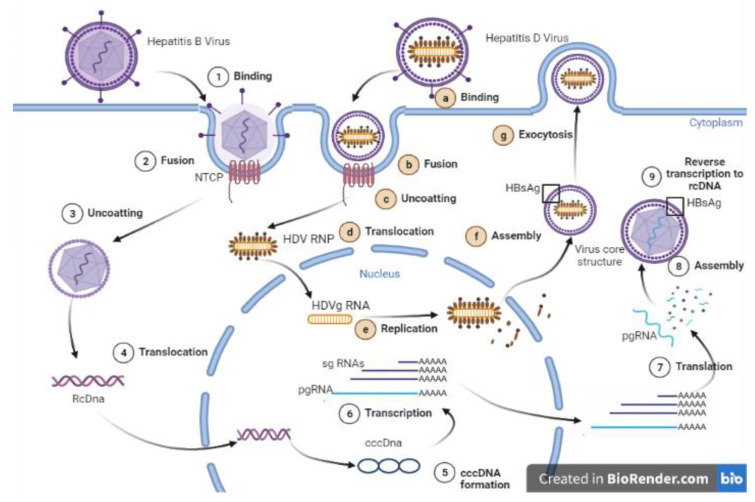
Upon interaction with the surface of the hepatocytes heparan sulphate proteoglycans (HSPGs), a region of HBsAg (large) binds to the receptor of HBV NTCP. For HDV, the viral RNP move (translocate) to the nucleus for transcription of genomic RNA first into complementary antigenomic RNA, then into new genomic RNA and mRNA using its host enzymes like polymerase II. The HDV ribonucoprotein (RNP) becomes envelope when it interacts with HBV, which then provide the surface protein (HBsAg) component. The viral and cellular components are assembled in a small vesicles and exocytosis. **Key:** Binding of HBV surface proteins (HBsAg) to the sodium taurocholate co-transporting polypeptide (NTCP) receptor (1,a), is initiating viral entry (2,b); After uncoating (3,c), HBV relaxed circular DNA (rcDNA) or HDV ribonucleoprotein complexe (RNP) is delivered to the nucleus (4,d); rcDNA is then repaired to form covalently closed circular DNA (cccDNA) and HDV genomic RNA (HDVg) is replicated (5,e); cccDNA is the template for transcription of viral RNA; cccDNA is transcripted to pre-genomic RNA (pgRNA) and sub-genomic RNAs (sgRNAs) (6); Viral mRNA is released from the nucleus and is translated to capsid particles and HDAg is produced and assembled (7,f); the pgRNA is then packaged into capsid particles (8); The pgRNA is reverse transcribed in the nucleocapsid (9); HDV viral particles binds to the hepatocyte membrane and are secreted (g).

**Table 1 microorganisms-10-02096-t001:** Serological, biochemical and molecular markers for HBV monoinfection and HBV/HDV coinfection.

Type of Markers	Diagnotic Markers	In HBV Mono-Infiection	In HBV-HDV Co-Infection
Serological markers	HB sAg	√	√
	Anti-HBs antibodies	√	√
	HBc Ab	√	√
	Anti-HDV antibodies	✕	√
	HB eAg	√	√
	Anti-HBe antibodies	√	√
	HBcrAg	√	√
Biochemical markers	AST	√	√
	ALT	√	√
	GGT	√	√
	AFP (associated with DCP)	If HBV-related HCC	If HBV- HDV related HCC
Mloecular marksers	HBV DNA	√	√
	HDV RNA	✕	√
	HBV RNA	√	√

Footnote: In green the markers used for first line diagnosis of HBV-HDV co-infection. AST and ALT are also applied in general biochemical screening for liver disease. In yellow, second line markers for HBV diagnosis. In red, molecular markers used after serological screening, for viral load determination and patient’s monitoring. In orange, biochemical markers used for detection of general liver disease (GGT) and HCC (AFP).

**Table 2 microorganisms-10-02096-t002:** The phases of the natural history and assessment of patients with chronic HBV infection as characterize by EASL-2017.

HBeAg Positive			HBeAg Negative	
	Chronic Infection	Chronic Hepatitis	Chronic Infection	Chronic Hepatitis
HBsAg	High	High or moderate	Low	Intermediate
HBeAg	Positive	Positive	Negative	Negative
HBV DNA	>log 7 IU/mL	log 4 to 7 IU/mL	>2000 IU/mL	>2000 IU/mL
ALT	Normal	Elevated	Normal	Elevated
Old Terminology	Immune tolerant	Immune reaction HBeAg positive	Inactve carrier	HBeAg negative chronic hepatitis

**Table 3 microorganisms-10-02096-t003:** Tools and markers that could be used for HBV and HDV diagnosis in resource limited countries.

	Biomarkers	Methods	Diagnotic Tools	Advantages	Limitations
**Serological markers**	HBsAg	ELISA	Architect quantitative HBsAg QT test	-HBsAg levels could predict response to treatment and SVR	-There is need for improved sensitivity, validation, and implementation in large cohorts.
			(Abbott laboratory)	-Levels could help determine loss or progression to liver disease in some cohorts	
			Elecsys quantitative test HBsAg II (Rochelaboratory)	-It is Cost-effective and available in some low- income contries.	
		CLIA	Liaison XL Murex quantitative test (DiaSorin laboratory)	-The turn-around-time is shorter compared to some HBV DNA assay	
		POC	Determine HBsAg (Alere laboratory)	POC	-It may miss some genotypes, especially mutations associated with impaired HBsAg release (genotype G).
				-Easy to use and cheap	
				-For scale up screening	
			VIKIA HBsAg (Biomerieux laboratory)	-Accessible for rural communities	
			DRW-HBsAg v.2 (Diagnostics for the real-world laboratory)	-Long self-life	
	HBeAg	ELISA	Architect quantitative HBeAg QT (Abbott laboratory)	-Can be used as a surrogate for HBV DNA levels in the absence of HBV DNA (for pregnant women) [116]-	Poor sensitivity and specificity because of cross-reaction with HBcAg.
			Elecsys quantitative HBeAg (Roche laboratory)	-Loss of HBeAg signifies seroconversion to anti-HBe is a current treatment end point	
		Immunoassays	CLEIA Limpulse G600II HBcrAg (Fujirebio laboratory)	-Could distinguish between active CHB carriers and HBeAg negative carriers.	-Many factors could result to inapropriate interpretation, such as anti-Hbe, mutations acffecting expression of HBeAg.
				-Correlates with cccDNA	It is not widely available
			iTACT-HBcrAg (Fujirebio laboratory)	-It can help categories cirrhotic or HCC risk patient who are HBeAg negative	-Further validation with number of cohorts from communities with high prevalence genotypes for developing countries
		POC	HBcrAg-RDT [39]	-Simple	Identify only highly viremic patients
				-Affordable	
				-Can be used in low- and middle-income countries	
	HBcAb	ELISA	Architect anti HBc II (Abbott laboratory	-HBc is a classical serologic HBV marker that has been clinically used for more than 35 years	-It should be combined with other quantifiable markers, such as HBV DNA and HBsAg.
			Lumipulse presto II (Fujirebio laboratory)	-Reflects the host immune response against HBV	
	Anti-HDV Ab	CLIA	Liaison XL Murex anti-HDV IgG and IgM (Diasorin laboratory)	-It can help categorising HDV disease phases	-There is need for further validation in more extensive populations.
		ELISA	ETI-AB DELTAK-2 (Diasorin laboratory)	-Great sensitivity and specificity. It identification of occult HBV (OBI)	
		POC	Prototype [113]	-Cost effective	-The Wantai assay has narrow range of quantification
				-Easy to use	
				-Can be used in hard-to-reach areas	
				-Validated	
**Biochemical markers**	ALT	Biochemical assay	Automatic analyser 7600 (Hitachi laboratory)	-Correlation of ALT with HBV DNA and HBcAb	-It has been reported that ALT levels may sometimes miss classified some percentage of patients with noticeable liver inflation [117].
				-Establishment of TREAT-B score for predicting treatment eligibility	-Further validation is required for this marker
	AFP	Immunoassay	Limpulse G1200 (Fujirebio laboratory)	-Diagnosis of HBV-related HCC	-Low sensitivity and high false positivity
					-Cannot distinguish between small HCC and Cirrhosis
**Molecular Markers**	HBV DNA	Real-time PCR	Real time HBV quantification kit (Abbott laboratory)	-High sensitivity	-Costly
				-Quantitative	-Unavailable in developing countries
		Recent nucleotide Assays	Aptima HBV Quant Assay (Hologic laboratory)	Is commonly used in Europe	-Still inaccessible to developing countries.
			GeneXpert HBV viral load (Cepheid laboratory)	Used for different viral diagnosis (HIV, HCV, HBV etc). The turn-around time is short (within 1 hour) to obtain result	-It requires large volume of serum samples (0.6 to 1.0 mL)
			Lamp (New England Biolabs)	-Cost-effective	-There need for further field evaluations.
				-A new solution for resource limited countries	-Further validation data on HBV genotypes in the resource- limited countries is needed [65].
	HDV RNA	PCR for genotyping	Nested PCR	-Validated for HDV Genotyping	-Prone to contamination when care is not taken
				-More sensitive than usual PCR	
				-Cheaper than qPCR for HDV RNA-	It could be time consuming

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
