# Peer review of "Viral Diagnosis of Hepatitis B and Delta: What We Know and What Is Still Required? Specific Focus on Low- and Middle-Income Countries"

_microorganisms, 2022, doi:10.3390/microorganisms10112096_

Round 1

Reviewer 1 Report

This review is a good manuscript. However, the focus is not strong, and the conclusions do not suggest how and what to do in low - middle-income countries.

Author Response

Point 1: This review is a good manuscript. However, the focus is not strong, and the conclusions do not suggest how and what to do in low – and middle-income countries.

Response 1. We thank you for your insightful suggestions. In the revised version, we have included a new « summary » section and a new Table 3, where we provide suggestions on how some of the tools described in the manuscript could be implemented in resource-limited settings

Reviewer 2 Report

The review provides an insight of both existing and underdevelopment diagnosis tools for HBV and HDV. Generally the review is well-written, very low errors are detected. 

- The authors need to highlight the contribution of this review to the field especially that many previous studies and reviews dealing with this topic have already been published. 

- The abbreviations in the abstract and introduction sections should be written in full for the first time. The authors don't need to mention the details of them after then. 

- In HBeAg, please indicate what e refers to.

- The authors need to incorporate more recent references following 2020. Many of cited references are in the peroid 2011-2018.

- The table title should be placed above the table (not below) - Th font size of table 2 need to be increased to be more clear to reader. - The figures legends should be more descriptive and informative to the reader. Herein, the authors just define the abbreviations represented in the figure.

Author Response

Responses to Reviewer 2 comments

Comments and Suggestions for Authors 

The review provides an insight of both existing and underdevelopment diagnosis tools for HBV and HDV. Generally, the review is well-written, very low errors are detected.  

Point 1: The authors need to highlight the contribution of this review to the field especially that many previous studies and reviews dealing with this topic have already been published.

Response 1: We thank you for the comment.

And indeed, we did emphasize in the text the specificities of our review that focused on the use of simplified diagnostic tools for low- and middle-income countries. On this point, we have also provided a summary paragraph at the end of the manuscript detailing the options available for some of the tools.

Point 2: The abbreviations in the abstract and introduction sections should be written in full for the first time. The authors don't need to mention the details of them after then.  

Response 2. We thank the reviewer for the wonderful observation. We have inserted all abbreviations in full in the abstract and introduction and avoided repetitions.

  Point 3: In HBeAg, please indicate what e refers to.

Response 3: We thank the reviewer for the comment. Indeed, the origin of the “e” denomination for this antigen is quite unclear. It seems that this appellation stemmed from the listing of the antigenic diversities of the Australian antigen (HBsAg) in the 1970s, being the “e” antigen the fifth to have been found. It will be only after that it was discovered that the e-antigen behaved biophysically more like a soluble protein, and was not a component of an HBV-related particle, the virion, the subviral HBsAg particles, or the capsid/core (HBcAg). Based on this data, Lars Magnius coined the term extra particulate for the e-antigen specificity in 1975. However, since the origin of the « e » denomination still seems obscure, we prefer not to enter into the details of it and added the reference from Magnius et al. in the text (Magnius LO (1975) Characterization of a new antigen-antibody system associated with hepatitis B. Clin Exp Immuno- 20(2):209–216), (reference number 113).

Point 4: The authors need to incorporate more recent references following 2020. Many of cited references are from the period 2011-2018. The papers cited in the review maybe a reflection of the period of new diagnostic tool development (2018 to 2020) because of that our search did not yield papers that are more recent than 2021.

Response 4. We thank the reviewer for the comments, and we apologize for missing some of the latest references. We have added five publications on the topic from 2021 to 2022, two for 2022 and three for 2021 respectively. The papers were assigned reference numbers 1111, 1114, 1115, 1116, and 1117.

Point 5: The table title should be placed above the table (not below) - The font size of table 2 need to be increased to be clearer to reader.   

 Response 5: Thank you for this observation. We have placed the title of the table above the table and increased the font sizes to 18 and 14 in the ppt.

Point 6: The figure's legends should be more descriptive and informative to the reader. 

 Herein, the authors just define the abbreviations represented in the figure. 

Response 6: Thank you for this comment. The legends of the figures were described and explained to reflect the content of the figure as suggested. The definition of abbreviations is given as a footnote.

Reviewer 3 Report

.

Author Response

Reviewer 3 comments

Comments and Suggestions for Authors

Response: We would like to thank the reviewer for reading our manuscript and for the language editing suggestion. We would like to use the English language services of the publisher for this comment.

Reviewer 4 Report

In this review, Ceesay and colleagues describe the current challenges in achiving HBV eradication by 2030, with a special emphasis on limitations faced by developing countries in particular.

The review is presented in a logical manner and provided adequate detail. I gained a good basic understanding of thr diagnostic tools available.

Overall, this was an interesting review and a well-organized manuscript. I have no serious concerns or any major issues with the manuscript in general.

However, some concerns need to be addressed.

·       On line 51: The authors mention Figures and Tables throughout the manuscript but no figure/ tables are included.

·       Line 57/58: The term ‘contamination’ is a little inaccurate. Saying “exposed to HDV at the same time” sounds more suitable for the context.

It would be extremely useful for readers if the authors can include a figure to summarize which biomarkers are useful at the different stages of hepatitis infection. Though the authors describe this throughout the text a figure would be especially useful for non-specialists.  

There are some formatting and font inconsistencies that need to be corrected.

Author Response

Responses to Reviewer 4 comments

Comments and Suggestions for Authors

In this review, Ceesay and colleagues describe the current challenges in achieving HBV eradication by 2030, with a special emphasis on limitations faced by developing countries in particular. The review is presented in a logical manner and provided adequate detail. I gained a good basic understanding of the diagnostic tools available.

Overall, this was an interesting review and a well-organized manuscript. I have no serious concerns or any major issues with the manuscript in general.

However, some concerns need to be addressed.

   Point 1: On line 51: The authors mention Figures and Tables throughout the manuscript but no figure/ tables are included.

Response 1: We would like to thank the reviewer for the wonderful remark and we apologize for the inconvenience. Indeed, the tables and figures were sent in a separate file from the manuscript due to formatting requirements. We hope that the reviewer will have access to the Figures and Tables of this revised version, otherwise, we invite the editorial managers to ask us for a different manuscript outline.

   Point 2: Line 57/58: The term ‘contamination’ is a little inaccurate. Saying “exposed to HDV at the same time” sounds more suitable for the context.

Response 2: We thank the reviewer for the suggestion and we corrected the sentence accordingly.

Point 3: It would be extremely useful for readers if the authors can include a figure to summarize which biomarkers are useful at the different stages of hepatitis infection. Though the authors describe this throughout the text a figure would be especially useful for non-specialists.  

Response 3. We thank the reviewer for the insightful comment. We have included a figure (Figure 3) to show different biomarkers at various disease stages for both HBV mono and HBV-HDV co-infection.

Point 4: There are some formatting and font inconsistencies that need to be corrected.

Response 4: We thank the reviewer for this keen observation. We have harmonized the font size throughout the text.

Round 2

Reviewer 3 Report

None